# ‘Next Stop: Mum’: Evaluation of a Postpartum Depression Prevention Strategy in Poland

**DOI:** 10.3390/ijerph191811731

**Published:** 2022-09-17

**Authors:** Magdalena Chrzan-Dętkoś, Natalia Murawska, Tamara Walczak-Kozłowska

**Affiliations:** Institute of Psychology, Faculty of Social Sciences, University of Gdansk, 80-309 Gdańsk, Poland

**Keywords:** longitudinal program evaluation, training for health professionals, the role of midwives, counselling support, online screening, postpartum depression

## Abstract

In the article we present a mid-point evaluation of the postpartum depression (PPD) prevention strategy in Poland. As PPD is associated with potential negative consequences for the mother and infant, the need to introduce screening and treatment is vital. The project covered over 21,500 women in the first year postpartum. The average score in the Edinburgh Postnatal Depression Scale (EPDS), in a screening provided in direct contact, was 4.73 (*SD* = 4.14, n = 7222), and increased in 55% of women in the follow-up study. In online screening the average score in the EPDS assessment was 16.05 (*SD* = 5.975, n = 10,454). The ‘probable depression’ rate (EPDS > 12) in ‘direct’ contact is 7.3%, and on the online platform—77%. Additionally, 26% of possibly affected mothers assessed in ‘direct’ contact benefited from psychological consultations. The average score in the EPDS among mothers who benefitted from consultations is 16.24 (*SD* =4.674, n = 231). Approx. 82% of healthcare providers raised their knowledge of PPD after training sessions. Maintaining the assumptions of the program: training for medical staff, screening conducted throughout the first twelve months postpartum, online platform with the possibility of self-screening and early psychological intervention seem to be justified actions, leading to a higher number of women with risk of PPD referred.

## 1. Introduction

The perinatal period (from pregnancy to 1 year after childbirth) is a vulnerable time for the onset and recurrence of mental disorders, with perinatal depression, anxiety disorders, and post-traumatic stress disorder (PTSD) as the leading diagnoses. PPD can be found in 17.22% of the world’s population [1], and approximately 1 in 5 women will develop a perinatal mental disorder [2,3,4,5,6,7]. 

If PPD is left untreated, the symptoms last seven months on average, but they can extend into the second year after giving birth [5]. Unfortunately, despite common knowledge about PPD, only 30.8% of affected women are identified in clinical settings, 15.8% receive treatment, and 6.3% receive adequate treatment [8]. In addition, the data from American Maternal Behavioral Health Policy Evaluation (MAPLE) study reveals that the prevalence of suicidal thoughts or self-harm in the years pre- and post-pregnancy almost tripled between 2006 and 2017, from 0.2 to 0.6 [9]. It shows the critical need for prevention and early intervention to help affected mothers.

Additionally, maternal depression in the perinatal period raises the risk for an adverse outcome in all child measures [10,11]. For example, maternal prenatal depression doubles the risk of premature labor [12,13,14]. It also influences a child’s long-term emotional and social development. The Avon Longitudinal Study of Parents and Children (ALSPAC) population study (n = 9848) showed that postpartum depression had negative and lasting developmental consequences also seen in children at 18 years of age [10]. Particularly sensitive groups of children are those whose mothers suffered from depression between their second and eighth month of life. These children were four times more likely to have problems with behavior between the third and fourth years of life. They have a two times higher risk of math problems at 16 years old and have a seven times higher risk of depression at 18. On the other hand, their mothers were much more likely to have depression 11 years after the occurrence of postpartum depression.

Many countries implement prevention programs by recognizing the emotional and economic burden [4] that postpartum depression imposes on families and society [15]. “Beyondblue” in Australia is well described and presented in the literature programme [15,16,17]. Reilly et al. [15] meta-analysis describes available national or regional perinatal depression screening program, showing that they increase referral rates and service use and are associated with more optimal emotional health outcomes.

Existing in most countries, the practice of midwives’ postpartum home visits and control visits at the pediatrician’s office creates a favorable circumstance for the provision of screening. For example, in Poland, midwives and nurses contact new mothers at least once a month during the prenatal period and a few times in the postpartum period. The infant immunization schedule also allows direct contact with a mother in the first year postpartum. However, in Poland, until 2019, there were no systemic solutions for the early diagnosis and treatment of mental disorders in the perinatal period. The latest extensive epidemiological data on perinatal mental disorders in Poland are available for 2005. Rymaszewska et al. [18] reported that PPD affects approximately 10–20% of Polish women. A smaller study by Jaeschke et al. [19] indicated that out of 434 assessed Polish women, 15.2% scored ≥ 13 points on the Edinburgh Postnatal Depression Scale (EPDS), thus fulfilling the screening criteria for PPD. Using the data provided by the Polish Statistical Council [20], it can be estimated that either 47,667 (assuming that 13% of women in the population are affected) or 73,334 (assuming that 20% of women in the population are affected) females in Poland suffer from PPD. The situation in Poland changed when in 2019, a new standard of perinatal care (Regulation of the Minister of Health) imposed the obligation to monitor depressive symptoms during pregnancy and in the postpartum period on the healthcare providers. It is a huge step forward. However, there is still a lack of paths to quick referral and low awareness of the problem of maternal mental health. The standard waiting time for a visit to a psychiatrist or psychotherapist covered by the National Health Fund in Poland ranges from 4 weeks to one year in large cities in the northern region of Poland (information available here: https://terminyleczenia.nfz.gov.pl, accessed 24 July 2022). Moreover, many people living in rural areas have limited access to mental health professionals, who usually see patients in larger cities.

### 1.1. Pilot National Program

The ‘Next Stop: Mum’ PPD preventive program, covering the northern region of Poland (with 5,833,595 inhabitants in total), is a part of the project: ‘Development of the concept and substantive assumptions of health policy programs to be implemented under the competition procedure’ no. POWR.05.01.00-IP.05-00-006/18 and co-financed by the EU Society Funds under Operational Program: Knowledge Education Development. The realization was set for 2019–2023. The program’s main objective is to increase the early detection of postpartum depression symptoms through education and public awareness of PPD. The project is managed by a consortium established by the Copernicus Podmiot Leczniczy sp z o.o, University of Gdańsk, Poland and Fundacja Twórczych Kobiet. This project is still in progress. The end is planned on April 2023. This midpoint evaluation of the implementation process had a practical component—it aimed to help plan further steps and potential changes that could optimize the outcomes or modify the assumption of the project.

#### The Aim of the Study

This study aimed to describe and evaluate the implementation process of a postpartum depression prevention project in Poland’s primary healthcare setting: assess the number of potentially affected women in the direct and online assessment of the risk of PPD, the number of psychological consultations and the improvement of knowledge among the medical staff.

## 2. Materials and Methods

### 2.1. Study Design

The research study was based on the ‘Next Stop: Mum’ PPD preventive program. The authors of the program divided its actions into four main areas.

#### 2.1.1. Education of Health Care Professionals (Mainly Midwives, Nurses, and Physicians in Primary Healthcare Settings)

The ‘Next Stop: Mum’ project covers 37 primary healthcare centers and 7 state hospitals. Medical staff working from the collaborating centers and hospitals obtain 6-h training concerning mental health in the perinatal period and screening methods. Psychologist trainers conduct training in small groups in a workshop formula (the brief theoretical introduction is supplemented by numerous case studies and role-plays). It ends with a test verifying the acquired knowledge and skills. We also provide supervision for healthcare professionals who collaborate on the project.

#### 2.1.2. Educational and Information Campaigns 

Patients using the care of cooperating primary health centers and hospitals receive leaflets containing information on mental health in the postpartum period. One early task was the project’s website (https://przystanekmama.copernicus.gda.pl, accessed on 24 July 2022) summarizing information on the PPD. A vital element of the website is an online version of the Edinburgh Postnatal Depression Scale, which enables anonymous self-screening and the possibility of receiving immediate feedback (with suggestions for further steps to be taken, e.g., contact with a psychologist). In addition, women can contact psychologists through the direct message function on the project’s Facebook and Instagram accounts.

#### 2.1.3. Screening and Follow Up

Screening for PPD is provided with EPDS. Most screening assessments are conducted in hospitals, during postpartum home visits, visits for immunization schedules or standard medical appointments. Each time, the woman is asked for written consent to participate in the study.

Women can also complete anonymous self-screening for PPD on the project’s website. Although the program is designed for women up to 12 months after childbirth, in the case of the online platform, people with older children can also use it. Similarly, there are no gender restrictions in the survey. Men can also complete the survey on our page at any time. Interestingly, about 2.5% of completed online surveys come from men. Nevertheless, the project’s assumptions and the activities carried out within them are addressed to women in the postpartum period.

For the implementation of the project (2019–2023), we have set an indicator of 36,000 screening assessments with 6000 online self-screening examinations included. For April 2022, we had 11,238 completed EPDS scales in direct assessments and 10,454 completed self-screening online.

During the follow-up, we randomly chose women between the third and tenth months postpartum to re-assess the intensity of depressive symptoms with the EPDS. We also ask them about the details of the screening procedure conducted by the midwife/nurse to monitor its correctness. This stage of the study is provided via telephone conversation.

#### 2.1.4. Consultations with Psychologists

Women who obtain results defined as ‘probable depression’ (12 points or above) or subclinical, possible PDD (10–11 points) results in the EPDS assessment have an opportunity to participate in free-of-cost three psychological consultations. Consultations take place in primary healthcare centers or state hospitals and can be conducted remotely.

For the implementation of the project (2019–2023), we have set an indicator of 2100 consultations; for April 2022, we have provided almost 650 consultations—300 women benefited from the appointment with psychologists. In addition, the data of 231 women have already undergone administrative procedures (stamping, preparation of payroll for contractors) and have been entered into the database conduct. Approx. 90% of all consultations were provided remotely (mainly via Skype and messenger).

A summary of the intervention path and the procedure of the study is presented in Figure 1.

### 2.2. Setting

PPD preventive program based in Northern Poland is open to women in a postpartum period whose child is alive and who live in one of three northern voivodeships.

### 2.3. Measurement Tools

#### 2.3.1. Edinburgh Postnatal Depression Scale

The intensity of the PPD symptoms is assessed with the Edinburgh Postnatal Depression Scale [19], Polish version: [20]. The 10-item EPDS is the most commonly used depression screening tool in perinatal care. In the United Kingdom, the National Institute for Health and Care Excellence (NICE) guidelines and The United States Preventive Services Taskforce recommend screening pregnant and postpartum women with the EPDS [21,22].

Ten questions enable us to obtain up to 30 points (the higher the result, the more increased symptoms of postpartum depression). The positive predictive value of EPDS is estimated to be 70% [19] or even 90% [23]. The cut-off values of 10 or higher and 13 or higher are most often used to identify women who might have depression [23].

#### 2.3.2. Sociodemographic Survey

In the online and direct screening assessments provided by midwives and nurses, we use a short sociodemographic survey regarding age, the number of children, perceived economic and social status, and primary data on a newborns’ and mothers’ conditions after childbirth.

#### 2.3.3. The Modified Version of the Test of Antenatal and Postpartum Depression Knowledge

The Test of Antenatal and Postpartum Depression Knowledge by Jones et al. [16,17] measures the medical staff’s knowledge of the onset, prevalence, comorbidity, symptoms, associated risk factors, assessment, and treatment strategies of antenatal depression and PPD. The original version consists of 20 items; however, we slightly modified this test by adding a few additional questions concerning the program and method of calculating and interpreting the results of the EPDS scale.

#### 2.3.4. Psychological Consultation Cards

For every woman who benefited from the psychological consultations we obtain anonymized information about EPDS scores, depressive symptom checklist assessment, sociodemographic data, information about the earlier treatment of mental health problems, and the character of the consultation (intervention, psychoeducation, diagnosis, elaboration of further treatment and support).

### 2.4. Compliance with Ethical Standards

#### Ethics Approval and Consent to Participate

All ethical protocols for human research were received. The Ethics Board approved the study protocol for Research Projects at the Institute of Psychology, University of Gdańsk (decision no. 20/2019). The research was conducted following The Code of Ethics of the World Medical Association (Declaration of Helsinki). All study participants provided informed written consent prior to study enrollment. All participants were informed about the project’s purpose, data collection and processing, and the possibility of using the collected data for subsequent statistics.

## 3. Results

### 3.1. Screening for PPD

Among those 11,238 questionnaires completed in direct assessments, 7345 have already undergone administrative procedures (stamping, payroll preparation for contractors) and have been entered into the database to conduct a telephone follow-up. The average age of participants was 30.15 years (*SD* = 5.11). Further, 66.6% of the surveyed women were married (n = 484), 30.3%—in an informal relationship (n = 2190), 1.5% (n = 106) were independent mothers. Additionally, 4190 women had higher education (58%), 1999 secondary (27.6%), 583—basic vocational (8.1%), and 167 primary education (2.3%). Moreover, 51% of women (n = 3688) lived in a city with more than 250,000 inhabitants, 5.2% (n = 376)—in a city with less than 250,000 inhabitants, 6% (n = 442) in a city with up to 100,000 inhabitants, 11.8% (n = 856) in the city with up to 50,000 inhabitants. Next, 23.7% lived in the countryside (n = 1717). 30.7% had a caesarean section (n = 2217). The children were born at 39 weeks of gestation (*SD* = 2445). Finally, 163 mothers (2.3%) experienced postpartum health complications. 6% of examined mothers (n = 423) suffered from depression before childbirth, 2.6% (n = 191) from other mood disorders. There were no significant differences in the sex of the newborn infant of participating mothers (n = 3566 boys and n = 3518 girls).

Women (n = 7345) screened for depression by a midwife or a nurse had a low mean EPDS score of 4.73 (*SD* = 4.15). The detailed data on the occurrence of scores obtained from indirect assessments are presented in Table 1. One-way ANOVA revealed no significant differences between screening time points (*F* = 1.75, *p* = 0.105).

Women (n = 10,454) who used the online platform to self-screen reported a significantly higher mean EPDS score of 16.05 (*SD* = 5.975). Moreover, we were surprised that men (n = 269) also filled in EPDS forms. They obtained 13.50 points on average (*SD* = 6.67, Min = 0, Max = 30).

The detailed data on the occurrence of scores obtained by women in online self-screening is presented in Table 2. Despite the information on the website that the study is for women in the first year of a child’s life, 1431 women with children of toddler age and older (up to 48 months) participated in the study.

We used one-way ANOVA to verify the differences between the mean EPDS scores obtained in different abovementioned moments of the assessments. The results for which the assessment date was missing (n = 1414) were not included in the analysis. The analysis revealed a significant difference between different screening time points (*F* = 7.191, *p* < 0.001). Subsequent post-hoc analyses with Tukey’s HSD tests revealed that the mean scores obtained by women 6–12 (*M* = 15.16) and 13–24 (*M* = 15.48) months postpartum in online self-screening were significantly lower than the mean scores obtained by women 0–1 month postpartum (16.79) and 1–2 months postpartum (16.57).

### 3.2. Follow-Up

Most women identified as having relatively high midwives’ competencies in communicating information about PPD (*M* = 5.94, *SD* = 6.45 assessment on a 0–7 point scale) and their ability to create comfortable conditions for the assessment and discussion of postpartum mental health. Direct follow-up contact at three to nine months after initial screening revealed increased mean EPDS scores for 55.5% of women. The detailed results are presented in Table 3. A significantly higher mean score was observed in follow-up screening (t = −12.11, *p* < 0.001).

The follow-up study also revealed a disturbing phenomenon: 19.6% (n = 254) of the women were not informed by the midwife about their results in postpartum depression screening. They were also not informed about the possibility of contacting a psychologist. In such a situation, we contact the midwife/nurse to remind them of the screening rules and discuss possible difficulties. In addition, midwives are encouraged to contact the cooperating psychologist to discuss any problems in communicating the result.

Moreover, we decided to assess whether there was a change in the category among women who in the original study had an EPDS total score within one of the three categories: normal range, slightly increased, or increased severity of PPD. The details of this analysis are presented in Table 4.

### 3.3. Education of Health Care Professionals

Among 323 participants who took part in training, 285 took part in the pre-and post-test. The analysis of the differences for dependent variables with the Wilcoxon test revealed that the results differed significantly (*Z* = −11.71, *p* < 0.001). Most healthcare providers’ improved knowledge of antenatal depression and PPD (78%, n = 232). However, among approx. 10% (n = 30), we observed lower scores than before training. Among approx. 12% (n = 36), we observed no difference between the two assessments.

It is worth mentioning that initially, participants had the most difficulty with the issues related to the risk factors for perinatal depression. Even after the training, many participants were still unaware of the possible consequence of untreated depression, which is, for example, suicide. On the other hand, they improved their understanding of the negative consequence for the infant, and they had quite a good understating of the antenatal and PPD symptoms and sufficient knowledge about best practices in the case of baby blue. The authors’ previous study [24], covering the year when the obligation of PPD screening was introduced, revealed insufficient knowledge about antenatal and postnatal depression among healthcare professionals. Training and continuous education are essential to ensure midwives’ competency in dealing with mental disorders.

### 3.4. Psychological Consultations

For the implementation of the project (2019–2023), we have set an indicator of 2100 consultations. For April 2022, we have provided 650 consultations—300 women benefited from the appointment with psychologists. Additionally, 26% of women with increased ESDP scores who participated in the midwife/nurse-led screening test (data as of March 30, 2022) benefited from consultations with a psychologist. The average score of the EPDS was 16.24 points (SD = 4.674). Dominant symptoms reported by the participants are: lower mood (93%; n = 211) anxiety (82%, n = 186), mood swings (74%, n = 169), inadequate guilt (61%, n = 139), irritability (53%, n = 120). Twenty-seven women (13.5% of the group) reported suicidal thoughts.

Moreover, 46% of mothers (n = 104) had used psychological intervention earlier, and 33% (n = 65)—psychiatric help. At the time of referral, 5% of mothers (n = 12) used pharmacotherapy.

Furthermore, 86% (n = 196) came for consultations alone, 13% (n = 30) with a child, one person with a partner, and 58% (n = 120) of women were referred for further treatment in the public health service.

According to the assumptions of the Ministry of Health, the three consultations should have mainly a diagnostic function. For 45% of women, consultations constitute an important opportunity to develop a further path of help and treatment. They were referred to psychological therapy and pharmacotherapy and received information about support groups for mothers. However, in a psychologist’s opinion, for 46% of participants, consultations had a form of crisis intervention, and for 96% they have psychoeducational component.

## 4. Discussion

This study aimed to describe and evaluate the implementation process of a postpartum depression prevention project in Poland’s primary healthcare setting. The program’s implementation analysis is necessary for its success; isolating risk factors and implementing corrective actions at the halfway point may influence the mother’s support and potentially the infant well-being.

### 4.1. PPD Screening Procedures

We encountered differences concerning the average EPDS scores in online self-screening and questionnaires filled in the presence of a midwife. Higher average results were obtained in self-screenings conducted on our online platform. Probably greater anonymity was associated with greater freedom in disclosing information about one’s mental state. In the literature, we found similar results: in Mule et al. study [25], about 20% of perinatal women do not fully disclose information at depression screening. Forder et al. [26] reported that 38.9% of 1597 Australian women were uncomfortable with an enquiry about depression or anxiety symptoms, which can also contribute to not revealing information about their true well-being. According to Mule et al. study [25], factors affecting disclosure of depression and screening include self-stigma, fear of being perceived as a bad mother and lack of trust in their healthcare provider. In online screening, the dilemmas mentioned earlier disappear. In addition, perhaps women experiencing depressive symptoms may be more motivated to seek information about PDD and more likely to survey the website [27,28].

As women differ in their screening preference and their feelings of trust toward the caregiver, informing about the anonymous online platform for PPD screening should be essential during the healthcare worker’s meeting with a mother.

The follow-up assessment revealed that depressive symptoms might, in many cases, persist after childbirth, and the negative mental state may worsen—especially in those women who initially showed a moderate severity of depressive symptoms. Based on this result, screening should be performed at several time points during the first year postpartum. Healthcare professionals should be aware of PPD symptoms even at the end of the first year after birth. The number of affected mothers of children in toddler and preschool age taking part in online assessment (up to 82.9% of participants), suggests that screening for depression should be also available for mothers of older children.

### 4.2. Education of Health Care Professionals

Although the training is associated with an increase in general knowledge, unfortunately, even after the training, several participants were still unaware of the possible consequence of untreated or severe depression, which is suicide. Monitoring the results allows us to emphasize these issues during further trainings.

Based on the literature [26], the key to the optimal implementation process is the medical staff’s knowledge and engagement in realizing the project’s tasks. Therefore, one of the changes that we have introduced as a result of this evaluation is sending email newsletters to primary healthcare facilities to keep their knowledge up-to-date and increase the involvement of healthcare professionals in PPD screening.

### 4.3. Psychological Consultations

A number of factors seem to influence the lower-than-expected number of psychological consultations. First of all, the lower-than-expected percentage of women with an elevated EPDS score was identified in “direct” screening. Based on the literature, the Ministry’s guidelines assumed that 15% of women would receive EPDS scores greater than 12 points. Our preliminary calculation concerning the number of consultations was based on this indicator. However, the overall percentage of women with elevated results from direct contact participating in psychological consultations is 26%. The majority of women with increased results have accomplished only online screening, and not all of them could participate in consultations—the only condition for participation was living in one of the three voivodeships in northern Poland. Additionally, many mothers are screened in the first two weeks after delivery, where ‘baby blues’ is a natural reaction. Our follow-up study revealed also that midwives and nurses do not always inform mothers (about 19% of the cases) about the screening result and the possibility of contacting the psychologist, which also could have influenced the number of referrals. The screening obligation is still relatively new [29]. We try to keep in touch with midwives who do not discuss the results with their patients and educate about the correct method of screening.

The literature review provided insights into how we could increase the number of referred women. According to Reilly et al. [30], the odds of receiving a referral were up to 16 times greater for women who were asked about their past and current mental health than for women who did not receive any mental health assessment. Therefore, we introduced this item to our healthcare professional’s routine conversation with the mother.

## 5. Conclusions

Analyzing the results in the middle of the implementation process revealed some surprising phenomena. Among the unexpected results were:(1)The discrepancy in EPDS results in face-to-face contact and on the online platform: 7.3% versus 77%.(2)A lower number of women achieved higher scores in face-to-face contact (7.3% versus 15% expected).(3)The online screening raises much more interest than expected: the number of women participating in the online survey turned out to be almost twice as high as we assumed when planning the project (in the halfway: 10,454; we expected 6000 in total). Therefore, as disclosing complete information about mental state in contact with the nurse/midwife may be influenced by self-stigma, the possibility of performing an anonymous self-examination with clear guidance on further treatment steps is recommended.(4)As the initial screening score provided in direct contact increased by 55% of women, it is justified and highly recommended to extend the PPD screening period (Polish recommendations apply only to the first six weeks postpartum).(5)The number of psychological consultations was lower than expected. However, with a lower rate of EPDS score > 12, the referral rate of 26% of women was almost twice as large as we assumed at the beginning of the project.(6)Although the training concerning perinatal mental health contributes to increased knowledge about PPD, the awareness of the seriousness of this disorder is still low.

Despite the relative success of the ‘Next Stop: Mum’ project, taking into account the consequences of postpartum depression, which affect both the mother and the baby, we still need both the continuation of such activities and the tightening and cooperation of psychologists and midwives to ensure both high-quality screening tests and effective early psychological intervention.

## Figures and Tables

**Figure 1 ijerph-19-11731-f001:**
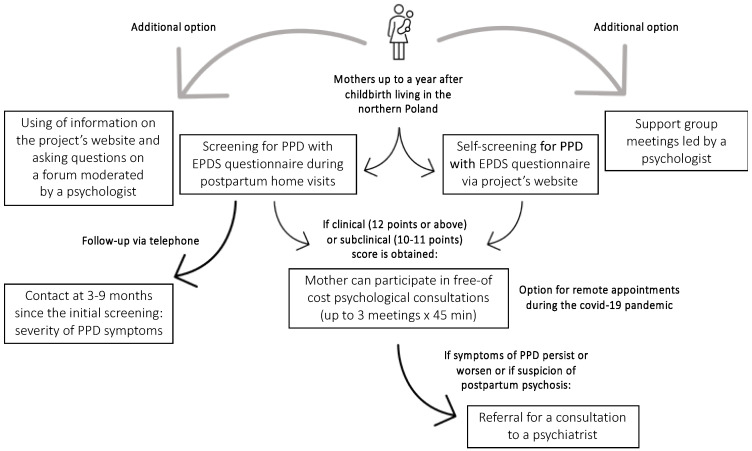
‘Next Stop: Mum’ project: the path of intervention and procedure of the study.

**Table 1 ijerph-19-11731-t001:** Occurrence of the scores obtained by postpartum women in direct screening for PPD with EPDS (regarding the cut-off points) in the ‘Next Stop: Mum’ project, n = 7345.

		Cut-Off Points		
The Moment of Assessment	Mean Score (*SD*)	Normal Range (0–9 Points)	Slightly Increased (10–11 Points)	Increased (12 Points and More)
Overall, n = 7345	4.73 (4.15)	86.5%	4.7%	7.3%
month postpartum, n = 5556	4.80 (4.11)	87.6%	5.0%	7.3%
1–2 months postpartum, n = 649	4.33 (3.92)	89.7%	4.3%	6.0%
2–3 months postpartum, n = 268	4.13 (4.19)	90.3%	3.7%	6.0%
3–4 months postpartum, n = 128	4.25 (4.34)	93.8%	1.6%	4.7%
4–5 months postpartum, n = 106	4.48 (4.15)	90.6%	3.8%	5.7%
5–6 months postpartum, n = 86	5.43 (5.56)	82.8%	3.4%	12.6%
6–12 months postpartum, n = 284	4.42 (4.20)	88.8%	3.2%	7.7%
No data provided, n = 268				

Note. The midwife provides screening mainly during a postpartum home visit or other occasions (immunization schedule, visits to the pediatrician’s office, etc.).

**Table 2 ijerph-19-11731-t002:** Occurrence of the scores obtained by postpartum women in an online self-screening for PPD with EPDS (regarding the cut-off points) in the ‘Next Stop: Mum’ project, n = 10,454.

		Cut-Off Points		
The Moment of Assessment	Mean Score (*SD*)	Normal Range (0–9 Points)	Slightly Increased (10–11 Points)	Increased (12 Points and More)
Overall, n = 10,454	16.05 (5.975)	14.7%	8.3%	77.0%
month postpartum, n = 694	16.79 (5.77)	12.0%	6.6%	81.4%
1–2 months postpartum, n = 472	16.57 (5.48)	11.4%	7.4%	81.1%
2–3 months postpartum, n = 350	16.48 (5.84)	10.6%	10.6%	78.9%
3–4 months postpartum, n = 309	15.88 (5.88)	16.2%	9.7%	74.1%
4–5 months postpartum, n = 280	16.70 (5.91)	13.9%	3.6%	82.5%
5–6 months postpartum, n = 192	15.71 (5.79)	14.9%	10.1%	75.0%
6–12 months postpartum, n = 867	15.16 (5.89)	17.9%	9.1%	73.0%
13–24 months postpartum, n = 718	15.48 (5.93)	16.4%	9.2%	74.4%
25–36 months postpartum, n = 328	16.27 (6.22)	14.9%	8.8%	76.2%
37–48 months postpartum, n = 385	17.14 (5.98)	11.4%	5.7%	82.9%

Note. Self-screening was conducted online via the ‘Next Stop: Mum’ project’s website: https://przystanekmama.copernicus.gda.pl (accessed on 24 July 2022).

**Table 3 ijerph-19-11731-t003:** Occurrence of the scores obtained by postpartum women in the follow-up screening for PPD with EPDS (regarding the cut-off points and change between the initial assessment and follow-up) in the ‘Next Stop: Mum’ project, n = 1297.

		EPDS’s Cut-Off Points			Change in the Severity of PPD Symptoms		
Time Since the First Examination	Mean Score (*SD*)	Normal Range (0–9 Points)	Slightly Increased (10–11 Points)	Increased (12 Points and More)	Increase in EPDS Scores	Decrease in EPDS Scores	No Difference in EPDS Scores
Overall, n = 1297	5.99 (5.02)	76.8%	6.7%	16.5%	55.5%	32.1%	14.1%
3 months after first screening, n = 85	5.55 (4.98)	76.4%	10.6%	12.9%	56.5%	29.4%	14.1%
4 months after first screening, n = 162	5.49 (5.10)	79.0%	4.9%	16.0%	54.3%	27.7%	17.9%
5 months after first screening, n = 143	6.76 (5.56)	70.6%	5.6%	23.7%	61.5%	28.7%	9.7%
6 months after first screening, n = 165	6.47 (5.36)	71.5%	7.2%	21.2%	60.0%	30.9%	9.1%
6 months after first screening, n = 120	6.51 (4.69)	78.3%	4.1%	17.5%	55.5%	28.3%	16.7%
8 months after first screening, n = 81	5.81 (4.58)	81.5%	8.6%	9.9%	58.0%	30.9%	11.1%
9 months after first screening, n = 112	6.84 (5.05)	69.6%	12.5%	17.9%	67.9%	21.4%	10.7%
10 months after first screening, n = 23	6.39 (4.83)	73.9%	0.0%	26.1%	65.2%	21.7%	13.0%

Note. Follow-up screening was provided by the Psychology graduate students from the Institute of Psychology, University of Gdańsk, Poland, and the SWPS University, Poland, via telephone conversation.

**Table 4 ijerph-19-11731-t004:** Change in the EPDS category (extracted based on the cut-off points) among women who took part in the follow-up and in the initial assessment provided by the midwife had an EPDS total score falling within one of the three categories: standard range, slightly increased, or increased severity of PPD; n = 1227.

	Follow-Up Assessment with EPDS ^1^		
Initial Assessment with EPDS Provided by a Midwife	Normal Range (0–9 Points),	Slightly Increased (10–11 Points),	Increased (12 Points and More),
Normal range (0–9 points)	72.3%	5.6%	13.5%
Slightly increased (10–11 points)	1.8%	1.3%	1.5%
Increased (12 points and more)	26.3%	3.4%	3.0%

Note: ^1^ Categories were extracted based on the EPDS’s cut-off points.

## Data Availability

Data sharing on demand to the authors.

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
