# Peer review of "‘Next Stop: Mum’: Evaluation of a Postpartum Depression Prevention Strategy in Poland"

_ijerph, 2022, doi:10.3390/ijerph191811731_

Round 1
Reviewer 1 Report
Thank you for the opportunity to review the manuscript titled 'Next Stop: Mum': Evaluation of a postpartum depression prevention strategy in Poland
This study followed 21500 mothers in their first year postpartum, specifically investigating postpartum depression (PPD). This paper targets a very important topic, however, there are major concerns.
In general, this study describes the incidence of PPD as well as intervention outcomes. However, the aims of this paper are not clearly presented. Referring to lines 79-80 does this paper actually describe a needs assessment?
English editing is required!
More specifically,
The introduction is lacking substantial information: What is the PPD incidence worldwide? Is the Polish incidence any different? What are the PPD risks for mothers, infants, and families? Are there any internationally published programs available on preventing and treating PPD.
Line 42 what is the burden (economic? Medical?)
Do lines 44-49 describe the existing screening programs mentioned in Line 43?
Lines 81-82 is not clear.
Please present the aim of the study
Methods – Study design (lines 84-86), Does this study test intervention outcomes?
Line 105 – Some important information is missing: When (relative to infant delivery) did the direct contact occur (months) and when did the follow-up occur (months)? When did the online screening (months) occur and if so when was the follow-up?
Figure 1 is not clear
Line 141- Please provide more information regarding the EPDS: what is its purpose? what is the standard cutoff?
Line 152 Is it a reliable and a valid tool? Did you test reliability of the modified version?
Line 165 – How was consent obtained from those who participated online?
It is not clear how long this study lasted? Table 1 indicates one year, where Table 2 reports longer duration. This must be indicated in methods.
Results: Line 171 – what does this mean " Among those 11 238 scales …"
It is not clear why …"screened for depression by a trained midwife… " line 174. The EPDS is a standard tool
In section 2.2 it seems that the inclusion criteria are presented (lines 138-9). Why did you report on participating men (line 203-4)? Are these men in the probable PPD range ? When were they screened (relative to their infant's birth)?
Table 4 – consider adding 'Total' in each of the rows and columns
Line 253- Please provide references. This statement has not been mentioned throughout the manuscript. Is suicide the only consequence of PPD, there may be others? (e.g., mother-infant attachment? abuse? Infant development?) Please elaborate (based on the literature) in the introduction.
Line 285 – Please refer to the literature, this may support your results.
Line 307 – This is confusing – this was not mentioned in the results section.
Line 309 – 'Do not always inform mother about the results…'
It is important to discuss this. Is this phenomenon known in literature? Is this ethical; was there no commitment to reporting the results if indicate a clinical issue?
Line 334 – 'puerperium period'? This is used here for the first time - worth clarifying
Author Response
Thank you very much for taking the time to read and provide a detailed review of our manuscript. It helped us improve it, and we hope it is now more precise. Please find below the detailed answers to your comments.
Point 1. The introduction lacks essential information: What is the PPD incidence worldwide? Is the Polish incidence any different? What are the PPD risks for mothers, infants, and families?
Thank you for this suggestion. We added these information to the manuscript.
Concerning Poland compared to other countries in terms of PPD prevalence, we did not know it before our study. According to Wang et al. (2021) meta-analysis, the prevalence of PPD varies in different countries. For example, in the United Kingdom, where the prevalence of PPD is estimated at 21.5%, – it is about 7,4%. While in New Zealand, another high-income country is 10.58%, PPD prevalence in Ghana and Egypt were 3% and 22.99%, respectively, while they belong to low-income countries, and their per capita GDP is similar. Our study shows that the prevalence of PPD is about 7,4 %.
Point 2. lacking information about internationally published programs available on preventing and treating.
Thank you for this suggestion: we added lacking information. We described cited metanalysis of Reilly et al. (2020), which discusses the programs described in the literature.
Point 3. Line 42 what is the burden (economic? Medical?) – t
The term "burden" was specified. We meant emotional and mental burdens for the new mothers.
Point 4. Do lines 44-49 describe the existing screening programs mentioned in Line 43?
Thank you for this comment. We specified it in the manuscript. First, however, we wanted to emphasize that the existing procedure: the care system for the mother and baby could be the base for implementing mental health screening.
Point 5. Lines 81-82 are not clear.
Thank you for this suggestion: the lines were corrected.
Point 6. Please present the aim of the study.
The aim of the study is presented more clearly.
Point 7. Methods – Study design (lines 84-86), Does this study test intervention outcomes?
We specified it in the corrected aim of the study. We examined the training effectiveness (knowledge test) and analyzed the records from psychological consultations. However, we only partially test intervention outcomes. We would not know, for example, if referral rates and public service use were higher after the program's implementation. As the screening was implemented only in 2019, we cannot compare for example, if more women were screened positively for PPD than before 2019.
Point 8. Line 105 – Some vital information is missing: When (relative to infant delivery) did the direct contact occur (months), and when did the follow-up occur (months)? When did the online screening (months) occur, and if so, when was the follow-up?
Thank you for this suggestion. We added the lacking information. Additionally, we describe the procedure below:
Direct study
Most screening assessments are conducted during postpartum home visits, visits for immunization schedules or standard medical appointments. Each time woman is asked for written consent to participate in the study. The direct screening may be conducted during the first year postpartum. Most screening is undertaken during the first three months of infant life; however, as pediatric nurses cooperate with us in the project, some mothers are screened in the second half of the first year of infants' life – during immunization visits.
Online study
Despite the information that the study is for women in the first year of a child's life, many women with children of an older age participated in the study. Nice guidelines recommend extending the screening period until the child is two years old - we decided to leave this data and put it in the article to show a possible significant tendency - the need for research in women with older children.
Point 9 Figure 1 is not clear.
Thank you for this comment. We wanted to present a graphic illustration of the assumption of the project. If Reviewer 1 finds it unclear, we can delete it.
Point 10. Line 141- Please provide more information regarding the EPDS: what is its purpose? What is the standard cutoff?
We provided the required information.
Point 11 Line 152 Is it a reliable and valid tool? Did you test the reliability of the modified version?
The reliability of the modified version was not tested. We added two questions concerning the knowledge of how to use EPDS, one checking the ability to score the reversed items and one concerning the reaction to 2 and 3 scores in item 10 asking about self-harm. This knowledge was essential for us to check if the nurses and midwives remembered that some questions in EPDS are reversed. Correctly calculating the Edinburgh Postnatal Depression Scale by midwives is very important to our project.
Point 12. Line 165 – How was consent obtained from those who participated online?
Participants who participate in online screening obtain information: "The collected data is anonymous and will be used for work related to the project's conduct and scientific research". They learn that they can stop the research at any time. If you have any questions, don't hesitate to get in touch with us via the website.
Point 13. It is not clear how long this study lasted. Table 1 indicates one year, where Table 2 reports a longer duration. This must be indicated in the methods.
Thank you very much for this comment. The program is designed for women up to 12 months after childbirth. However, in the case of the online platform, people with older children can also use it - that is why Table 2 presents data on mothers with older children. Information about this has been entered into the text.
Point 14 Results: Line 171 – what does this mean? "Among those 11 238 scales …."
Thank you for pointing out this phrase. I should have lived the word "questionnaires".
Point 15. It is not clear why …" screened for depression by a trained midwife…" line 174.
The EPDS is a standard tool
Thank you for pointing out this phrase. We changed it.
Point 16. In section 2.2, it seems that the inclusion criteria are presented (lines 138-9). Why did you report on participating men (lines 203-4)? Are these men in the probable PPD range? When were they screened (relative to their infant's birth)?
PPD occurs in approximately 8 to 10 per cent of fathers (Paulson, Bazemore, 2010; Cameron, Sedov, Tomfohr-Madsen,) with the highest prevalence within 3 to 6 months postpartum but with the possibility of developing over a year postpartum.
The men are not beneficiaries of the project, but they were able to complete the Edinburgh Postpartum Depression Scale questionnaire. In social media, which aims to promote the program, we drew attention to the risk of depression also by fathers. If in the opinion of the Reviewer, he enters unnecessary data, we can delete this information.
Point 17. Table 4 – consider adding 'Total' in each of the rows and columns
We do not think that this word is necessary in the mentioned table. If the reviewer thinks otherwise, we can change it.
Point 18. Line 253- Please provide references. This statement has not been mentioned throughout the manuscript.
The reference has been included.
Point 19. Is suicide the only consequence of PPD? There may be others . (e.g., mother-infant attachment? abuse? Infant development?) Please elaborate (based on the literature) in the introduction.
The consequences of PPD have been listed in the introduction.
Point 19. Line 285 – Please refer to the literature; this may support your results.
The reference has been included e.g Bucci et al.2019 and Naslund, 2016
Point Line 307 – This is confusing – this was not mentioned in the results section.
We have mentioned it in the introduction: description of the study, but to make this issue more clear, we have changed and elaborated the results concerning psychological consultations.
Point Line 309 – 'Do not always inform the mother about the results….'
It is important to discuss this. Is this phenomenon known in literature? Is this ethical; was there no commitment to reporting the results if they indicate a clinical issue?
I was wondering if this is a significant remark but I wonder whether the elaboration of this subject does not exceed the article's aim. Encouraging midwives to work on the project was not an easy task. When we receive information after the follow-up study that the midwife did not provide the result of the screening, we contact the midwife and educate them about the screening. Such an attitude to mental health results, unfortunately, reflects public awareness of maternal mental health. This is my hypothesis, but it may reflect some disregard for the well-being of young mothers. No midwife would do this with, e.g. cytology results. As a coordinator of this program and a University teacher, I believe this program is essential, as it helps educate midwives to pay attention to the emotional aspects of motherhood. In my opinion, midwives are the key to the program's success: educating them and paying attention to their understanding of the mother's situation can increase the number of referrals and effective screenings. However, the good cooperation is sometimes still a challenge..
Point Line 334 – 'puerperium period'? This is used here for the first time - worth clarifying.
Thank you for this comment, we have changed it.
Literature:
Bucci S, Schwannauer M, Berry N. The digital revolution and its impact on mental health care. Psychol Psychother. 2019 Jun;92(2):277-297. doi: 10.1111/papt.12222. Epub 2019 Mar 28. PMID: 30924316.
Naslund JA, Aschbrenner KA, Marsch LA, Bartels SJ. The future of mental health care: peer-to-peer support and social media. Epidemiol Psychiatr Sci. 2016 Apr;25(2):113-22. doi: 10.1017/S2045796015001067. Epub 2016 Jan 8. PMID: 26744309; PMCID: PMC4830464
Reilly N, Kingston D, Loxton D, Talcevska K, Austin MP (2020) A narrative review of studies addressing the clinical effectiveness of perinatal depression screening programs. Women and birth, 33(1), 51–59. https://doi.org/10.1016/j.wombi.2019.03.004
Wang Z, Liu J, Shuai H, Cai Z, Fu X, Liu Y, Xiao X, Zhang W, Krabbendam E, Liu S, Liu Z, Li Z, Yang BX. Mapping global prevalence of depression among postpartum women. Transl Psychiatry. 2021 Oct 20;11(1):543. Doi: 10.1038/s41398-021-01663-6.

Reviewer 2 Report
Dear authors,
I consider that you did a very nice and interesting paper. I have some comments on your document.
Line 17, please add after "Edinburgh Postnatal Depression Scale" the abbreviation (EPDS).
Line 20, It is not clear the value "on online platform - 77%.30%" please clarify.
Line 33, Please add the meaning of PTSD.
Lines 36-38, It is unclear if the 30.8% of women with PPD identified in clinical settings, only 15.8% receive treatment, and the 6.3% received adequate treatment, but what happens with the rest? I mean, with the 8.7%, could you explain?
Line 147, Sociodemographic survey, you have a perfect measure, but would you consider adding this parameter to the discussion?
Line 178, Table 1, please consider justifying why you are using the table values for "no data provided"
Line 215, Table 2, Your data about the moment of the assessment indicate an increase (upper 12 points) after 25 months postpartum. It could be nice to consider this data in your discussion in more detail.
Line 215, Table 2, why do you separate the months postpartum until the month 13? You used a scale in months 0-1, 1-2, 2-3, and you changed from 13-24, 25-36, 37-48. Could you please justify why you overlap some months?
Line 257, you mention the word "discrepancies" however, since they are different methodologies could be more correct to change the word for "differences"
Thank you for your effort and time
Author Response
Response to Reviewer 2
I consider that you did a very nice and interesting paper. I have some comments on your document.
Thank you very much for taking the time to read and provide a detailed review of our manuscript. It helped us improve it, and we hope it is now more precise. Please find below the detailed answers to your comments.
Point 1. Line 17, please add after "Edinburgh Postnatal Depression Scale" the abbreviation (EPDS).
The missing information was added.
Point 2. Line 20, It is not clear the value "on online platform - 77%.30%" please clarify.
The information we clarified, we hope that it is now more readable.
Point 3. Line 33, Please add the meaning of PTSD.
We added the missing information.
Point 4. Lines 36-38, It is unclear if the 30.8% of women with PPD identified in clinical settings, only 15.8% receive treatment, and the 6.3% received adequate treatment, but what happens with the rest? I mean, with the 8.7%, could you explain?
According to the authors (Cox et al., 2016), they do not receive treatment that is evidence-based or supported by guidelines.
Point 5. Line 147, Sociodemographic survey, you have a perfect measure, but would you consider adding this parameter to the discussion?
Thank you very much for this comment. We added information concerning the sociodemographic background of our participants.
Point 6. Line 178, Table 1, please consider justifying why you are using the table values for "no data provided"
We decided to exclude this information from the table.
Point 7. Line 215, Table 2, Your data about the moment of the assessment indicate an increase (upper 12 points) after 25 months postpartum. It could be nice to consider this data in your discussion in more detail.
Thank you very much for this comment. We added the sociodemographic data in the result section and mentioned them in the discussion. Unfortunately, however, we can't present more results in this article due to the limitations associated with the length of the manuscript.
Point 8. Line 215, Table 2, why do you separate the months postpartum until month 13? You used a scale in months 0-1, 1-2, 2-3, and you changed from 13-24, 25-36, 37-48. Could you please justify why you overlap some months?
We combined the months into groups (6-12, 13-24, 37-48) due to the much smaller number of respondents in these intervals and because we wanted to maintain comparability between groups. According to the Ministry of Health guidelines, our primary beneficiaries are women in the first year after giving birth.
Point 9. Line 257, you mention the word "discrepancies" however, since they are different methodologies could be more correct to change the word for "differences"
Thank you very much for this comment. We corrected this phrase.
Thank you for your effort and time
We thank you very much for your effort and time to read and provide a detailed review of our manuscript. It helped us improve it, and we hope it is now more precise.
